# Ca^2+^-Driven Selectivity of the Effect of the Cardiotonic Steroid Marinobufagenin on Rabbit Sinoatrial Node Function

**DOI:** 10.3390/cells12141881

**Published:** 2023-07-18

**Authors:** Sofia Segal, Yael Yaniv

**Affiliations:** Laboratory of Bioelectric and Bioenergetic Systems, Faculty of Biomedical Engineering, Technion-IIT, Haifa 3200003, Israel; sofi890@campus.technion.ac.il

**Keywords:** cardiac automaticity, calcium, glycoside, sodium

## Abstract

The synergy between Na^+^-K^+^ pumps, Na^+^-Ca^2+^ exchangers, membrane currents and the sarcoplasmic reticulum (SR) generates the coupled-clock system, which governs the spontaneous electrical activity of heart sinoatrial node cells (SANCs). Ca^2+^ mediates the degree of clock coupling via local Ca^2+^ release (LCR) from the SR and activation of cAMP/PKA signaling. Marinobufagenin (MBG) is a natural Na^+^-K^+^ pump inhibitor whose effect on SANCs has not been measured before. The following two hypotheses were tested to determine if and how MBG mediates between the Na^+^-K^+^ pump and spontaneous SAN activity: (i) MBG has a distinct effect on beat interval (BI) due to variable effects on LCR characteristics, and (ii) Ca^2+^ is an important mediator between MBG and SANC activity. Ca^2+^ transients were measured by confocal microscopy during application of increasing concentrations of MBG. To further support the hypothesis that Ca^2+^ mediates between MBG and SANC activity, Ca^2+^ was chelated by the addition of BAPTA. Dose response tests found that 100 nM MBG led to no change in BI in 6 SANCs (no BI change group), and to BI prolongation in 10 SANCs (BI change group). At the same concentration, the LCR period was prolonged in both groups, but more significantly in the BI change group. BAPTA-AM prolonged the BI in 12 SANCs. In the presence of BAPTA, 100 nM MBG had no effect on SANC BI or on the LCR period. In conclusion, the MBG effects on SANC function are mediated by the coupled clock system, and Ca^2+^ is an important regulator of these effects.

## 1. Introduction

The synergy between membrane currents, pumps and exchangers, together with the sarcoplasmic reticulum (SR), generates the coupled clock system that governs the spontaneous electrical activity of a heart pacemaker [1]. More specifically, spontaneous local Ca^2+^ releases (LCRs) from the SR activate Ca^2+^-sensitive mechanisms, such as Na^2+^-Ca^2+^ exchanger (NCX) and SK channels, and together with the funny current (I_f_) initiates action potentials of sinoatrial node (SAN) cells (SANCs). Ca^2+^ also activates adenylyl cyclase 1 and 8 (AC1 and AC8), and through the cAMP-PKA axis and the Ca^2+^-activated CaMKII axis, it also controls the degree of clock coupling [2].

Theoretically, the activity of the NCX is also controlled by Na^+^ concentrations. Because Na^+^ levels are tightly controlled by the Na^+^-K^+^ pump, Na^+^ and Ca^2+^ should have synergy between them. Previous works on SAN tissue showed that blockage of the Na^+^-K^+^ pump by digitalis affects SAN electrical activity [3,4]. Additional work in SANCs found changes in intracellular Ca^2+^ and LCR characteristics to be responsible for the biphasic effect of digitalis on SAN activity [5]. While these works have made a seminal contribution to understanding the synergy between the NCX exchanger and the Na^+^-K^+^ pump in response to drug perturbations, it remains to be determined if such synergy can exist under natural conditions.

In response to increased sodium loading, which is associated with an excessive salt intake diet or renal failure, marinobufagenin (MBG), a natural Na^+^-K^+^ pump inhibitor, is synthesized and secreted by the adrenal cortex [6]. Increases in MBG also serve as a biomarker of cardiovascular death, the majority of which are associated with some degree of SAN dysfunction [7]. The MBG effect (which is naturally produced in the body) on SANCs has not been measured before.

To investigate if and how MBG mediates between the Na^+^-K^+^ pump and spontaneous SAN activity, the following two hypotheses were tested: (i) MBG has a distinct effect on the beat interval (BI) due to a diverse effect on LCR characteristics, and (ii) Ca^2+^ is an important mediator between MBG and SANC activity.

To test these hypotheses, Ca^2+^ transients were measured with a confocal microscope during the application of increased concentrations of MBG on single spontaneously beating SANCs. In addition, this work studied the effect of Ca^2+^ chelation with BAPTA on the impact of MBG on SANC activity.

## 2. Materials and Methods

### 2.1. Animal Use

Animals were treated in accordance with the Technion Ethics Committee. The experimental protocols were approved by the Technion Animal Care and Use Committee (Ethics number IL-001-01-19).

### 2.2. SAN Cell Isolation and Culture

Single, spindle-shaped SANCs were isolated from the hearts of white New Zealand rabbits weighing 2.3–2.7 kg, as previously described [8,9]. Briefly, rabbits were sedated with ketamine and xylazine (0.1 mL/kg each) and anesthetized with sodium pentobarbital (200 mg/mL) diluted with heparin, administered through an intravenous cannula. After quick removal, the hearts were placed in a custom-made, silicone-covered chamber, and the SAN and right atrial tissues were separated, as previously described [10], while bathed in 37 °C Tyrode’s solution (125 mM NaCl, 5.6 mM KCl, 1.2 mM NaH_2_PO_4_, 24 mM NaHCO_3_, 5.6 mM glucose, 1 mM MgCl_2_ and 1.8 mM CaCl_2_, bubbled with 95% O_2_ and 5% CO_2_). Once separated from the surrounding atrial tissue, SAN tissue was processed as previously described [11] in order to obtain single spontaneously beating SANCs.

### 2.3. Ca^2+^ Imaging and Measurements

Ca^2+^ cycling in single spontaneously beating SANCs was recorded by monitoring the fluorescence of the Ca^2+^ indicator dye, Fluo-4 AM (Thermo Fisher Scientific, Waltham, MA, USA), using a LSM880 confocal microscope, as previously described [8]. SANCs were loaded with 5 μM Fluo-4 AM for 20 min, in the dark, at room temperature, and then washed with a 37 °C HEPES solution (140 mM NaCl, 5.4 mM KCl, 2 mM MgCl_2_, 5 mM HEPES, 1.8 mM CaCl_2_ and 10 mM glucose, titrated to pH 7.4 with NaOH). Fluorescence was observed by exciting the sample with a 488 nm argon laser and measuring fluorescence emission with LP 505 nm. Images of spontaneously beating SANCs were captured using a 40×/1.2 water immersion lens, using a line-scan mode (1.22 ms per scan; pixel size, 0.01 µm). Each cell was imaged before (control) and after drug administration. Line-scan images were analyzed using a modified version of a semi-automatic Matlab Graphic User Interface (GUI), the Sparkalyzer [11]. Experiments were performed on fresh cells less than 6 h after isolation. Ca^2+^ transients were semi-automatically detected, and Ca^2+^ sparks were manually marked.

### 2.4. Drugs

The cytosolic Ca^2+^ chelator BAPTA-AM was purchased from ENCO (Cayman chemical, MI, USA). Marinobufagenin, a natural Na^+^-K^+^ pump inhibitor, was purchased from BioTag (Cayman Chemical, MI, USA).

### 2.5. Statistics

Data are presented as mean ± SEM. Data were compared using a paired or unpaired student’s *t*-test. Differences were considered statistically significant at *p* ≤ 0.05 or *p* ≤ 0.01, as indicated in the respective figures.

## 3. Results

### 3.1. Selective Effect of MBG on Rabbit SANC Beat Interval

This set of experiments began with a characterization of the concentration response of rabbit SANC to MBG, as measured by BI. On treatment with 50 nM MBG, no change in BI was observed (Figure 1, Appendix A). On treatment with 100 nM MBG, 6 SANCs showed no change in BI (no BI change group), while 10 SANCs showed a prolonged BI (BI change group). At 1000 nM MBG, the BI was prolonged for both groups, with the prolongation only significant in the BI change group. Note that no significant difference in basal BI was observed between the no change and change group (Figure 1).

Further focus on the direct application of 100 nM MBG treatment, which led to the selective effect on BI, found that 8/20 SANCs belonged to the no-BI-change group, while 12/20 SANCs belonged to the BI-change group. Thus, the same proportion between affected and not affected cells was maintained whether a dose response or direct application of 100 nM MBG was used.

### 3.2. Selective Effect of MBG on Global and Local Ca^2+^ Parameters of Rabbit SANCs

Because MBG blocks the Na^+^-K^+^ pump, which indirectly blocks the NCX, and due to the effect of digitalis glycoside (a Na^+^-K^+^ pump blocker) on Ca^2+^ [5], the changes in Ca^2+^ were the mechanisms selected as a potential mediator. Figure 2A–D show representative examples of the effect of 100 nM MBG on the Ca^2+^ transient for the no-BI-change and BI-change groups. Global parameters (time to peak, 50% and 90% relaxation (T_50_ or T_90_, respectively)) did not shift from their basal values in either the no-BI-change (Figure 3A–C, Appendix A) or BI-change group (Figure 3D–F, Appendix A). Moreover, no change in the basal global Ca^2+^ parameters was observed between the groups (Appendix A).

Although the BI was maintained in the no-BI-change group, the LCR period was prolonged (Figure 4A), with an increase in the 50% LCR duration (Figure 4B). The LCR amplitude and LCR length did not change (Appendix A). In the BI-change group, parallel to the prolongation of the BI, the LCR period was prolonged (Figure 4C), as well as 50% LCR duration (Figure 4D), while the LCR amplitude and LCR length (Appendix A) did not change.

Note that although the prolongation of LCR periods was measured in both groups, a higher percentage of prolonged LCR periods was measured in the BI-change group (33.64 ± 3.1%) as compared to the no-BI-change group (23.31 ± 1.4%). After administration of 100 nM MBG, significant differences in the LCR period, spark length, 50% spark duration, normalized amplitude and amplitude difference were noted between the groups. Already at baseline levels, significant differences in the spark length, normalized amplitude and amplitude difference (Appendix A) were noted between the groups.

Following treatment with 50 nM MBG, no changes in BI or in the global Ca^2+^ parameters were measured (Appendix A). However, in the no-BI-change group, the 50% LCR duration increased with the prolongation of the LCR period. In the BI-change group, the LCR period became prolonged with decreases in the number of LCRs. At 1000 nM MBG, BI prolongation occurred in both groups, but it only reached significance in the BI-change group. In both groups, the 50% LCR duration increased and LCR period was prolonged following treatment with 1000 nM MBG.

### 3.3. Ca^2+^ Chelation Abolishes the Selective Effect of MBG

To verify that Ca^2+^ mediates the selective MBG effect on SANC function, SANCs were pretreated with BAPTA-AM, a Ca^2+^ chelator. A total of 10 µM of BAPTA-AM prolonged the BI in all 12/12 tested SANCs. Treatment of SANCs with BAPTA in combination with 100 nM MBG did not affect BI in all cells (Figure 5A). Figure 5B–D show representative examples of the effects of BAPTA-AM alone or in combination with 100 nM MBG. Appendix A shows that the global Ca^2+^ parameters (mean time to peak (Figure 6A), T_50_ (Figure 6B) or T_90_ (Figure 6C)) on treatment with BAPTA-AM combined with MBG were similar to those measured on treatment with BAPTA-AM alone. The LCR period (Figure 6D), LCR amplitude (Figure 6E) and 50% LCR duration (Figure 6F) were also similar across groups, while the LCR length (Figure 6G) was reduced in cells treated with BAPTA-AM and MBG compared to cells treated only with BAPTA-AM.

## 4. Discussion

At 50 nM MBG, no effect on BI was observed; however, changes in the LCR period were already measured in both groups. Thus, changes in Ca^2+^ are observed prior to changes in SAN BI. At 100 nM MBG, distinct differences in both LCR parameters and BI were observed between the groups. Na^+^-K^+^ pump blockade has been shown to increase diastolic and systolic Ca^2+^ levels in SANCs [5]. Based on the assumption that Na^+^-K^+^ pump blockade affects NCX, above a certain point, an increase in intracellular Ca^2+^ levels and Na^+^ leads to a decrease in NCX current and subsequently to a delay in action-potential ignition. A prolonged diastolic depolarization phase leads to a reduction in the amount of Ca^2+^ entering the cell per time unit, and consequently to a reduction in LCR release. In parallel, prolonged diastolic depolarization leads to a reduction in the funny current per time unit, resulting in reduced clock coupling, leading to a further reduction in Ca^2+^ activity and in phosphorylation activity [12,13]. In parallel, Na^+^-K^+^ pump blockade was shown to increase Src activity [14]. A PP2-driven reduction in Src activity leads to a decrease in the spontaneous BI in rat SAN due to an indirect effect on cAMP activity [15]. Thus, it is possible that an increase in Src activity in some cells compensates for the decrease in spontaneous BI through inhibition of the Na^+^-K^+^ pump. Thus, the MBG effect on SANC BI is rooted in a change in the degree of clock coupling. It is possible that different cells increased the Na^+^ levels due to varied sensitivities to MBG, thus leading to the diverse effect of MBG on BI. Intergroup differences in the baseline and post-MBG-treatment LCR period may support this hypothesis. Such a heterogeneity between cell groups was documented in intact SAN tissue in response to digitalis, a Na^+^-K^+^ pump blocker [3,4].

In the presence of a Ca^2+^ chelator, 100 nM MBG treatment did not have a distinct effect on BI in any SANCs. In all cells, 100 nM MBG does not affect the BI, and thus the BI-change group is eliminated. These observations further support the hypothesis that the coupled-clock system mediates the dual effect of MBG on BI. Ca^2+^ chelation disables the degree of clock coupling and prolongs the BI. When the coupled-clock function is reduced, no connection exists between the Na^+^-K^+^ pump and other clock mechanisms, and this therefore eliminates the effects of MBG on SANC function. Moreover, parallel to no change in BI, there was no change in the LCR period following treatment with MBG in the presence of a Ca^2+^ chelator. This further supports the hypothesis that Ca^2+^ is an important regulator of the MBG effect on SANC function and of the selective MBG effect.

In in vivo experiments, no change in heart rate was observed in MBG-treated rats [16]. A similar observation was reported in humans [17]. However, here we found a group of SANCs that prolonged their BI. It is possible that BI is prolonged in only some SANCs but not affected in the central SAN on treatment with MBG.

### Study Limitation

A biphasic effect of digitalis was shown before in SANCs [5]. It is possible that a lower concentration of MBG would lead to such an effect. However, this study focused on the concentration that led to a selective effect of MBG. Furthermore, the fact that MBG does not affect the heart rate in vivo implies that a biphasic effect may not occur in response to MBG.

The concentration used here is difficult to translate to clinically relevant concentrations. The fact that MBG does not affect the heart rate in vivo implies that the putative endogenous titers of the hormone are lower.

## Figures and Tables

**Figure 1 cells-12-01881-f001:**
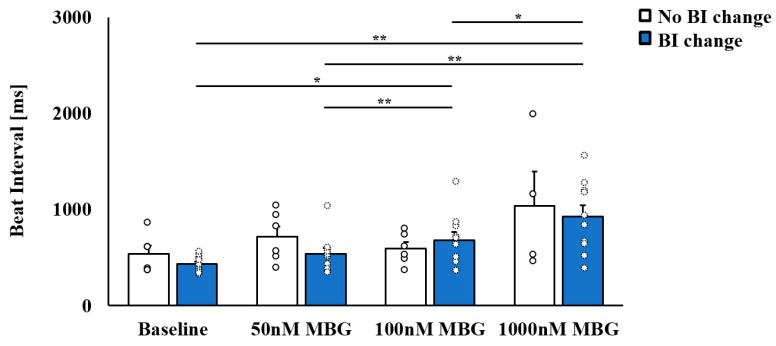
The selective effect of marinobufagenin (MBG) on rabbit sinoatrial node cell (SANC) beat interval (BI). Characterization of the dose response of rabbit SANC to MBG, as measured by BI. Upon treatment of SANCs with 50 nM MBG, no change in BI was observed. At 100 nM MBG, no change in BI was seen in some SANCs (no BI change group, *n* = 6), while prolonged BI was observed in others (BI change group, *n* = 10). Data are presented as mean ± SEM. * *p* ≤ 0.05, ** *p* ≤ 0.01.

**Figure 2 cells-12-01881-f002:**
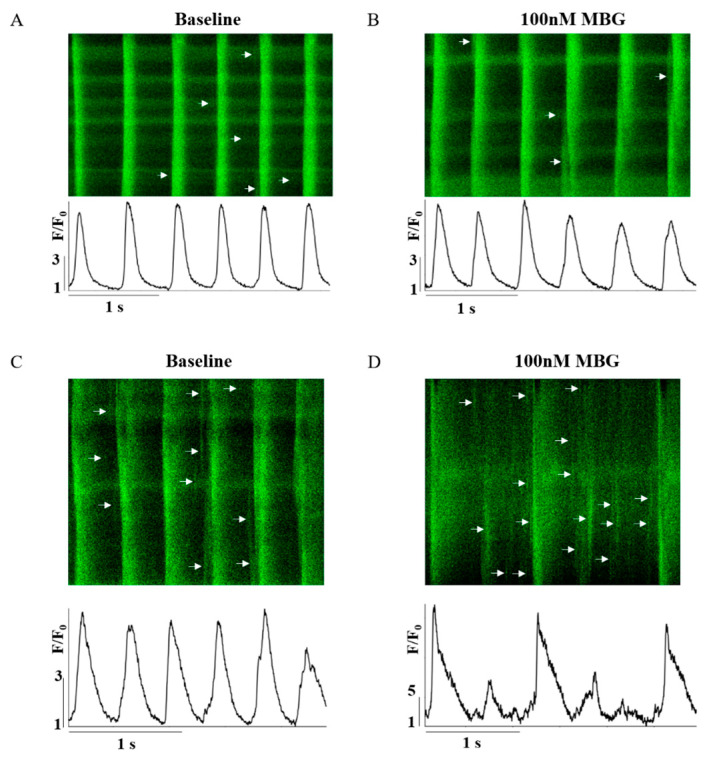
Representative example of the effect of 100 nM MBG on rabbit SANC Ca^2+^ transients. (**A**) Before and (**B**) after administration of 100 nM MBG to no-change-BI SANCs; and (**C**) before and (**D**) after administration of 100 nM MBG to change-BI SANCs.

**Figure 3 cells-12-01881-f003:**
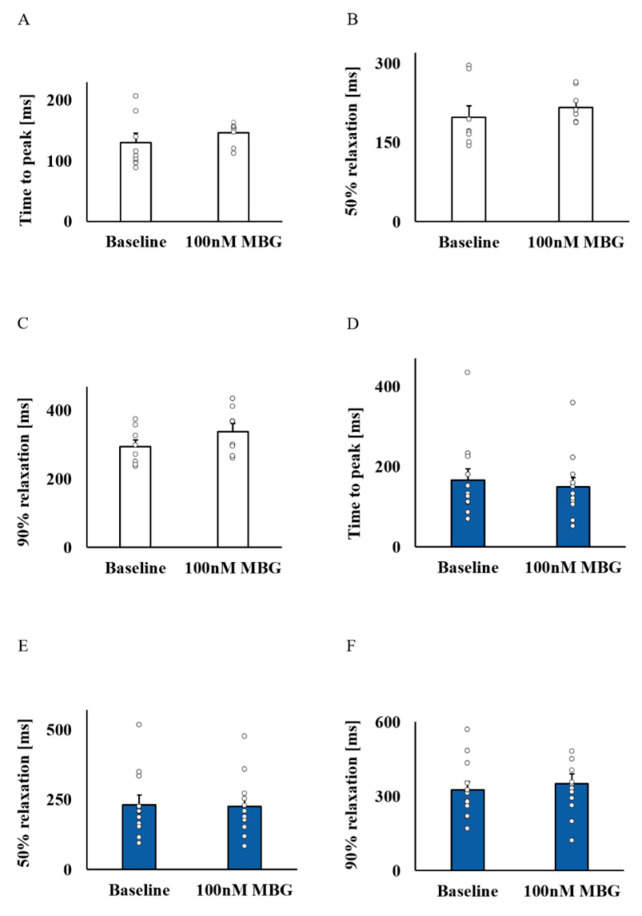
The effect of 100 nM MBG on rabbit SANC global Ca^2+^ parameters. For the no-BI-change group (*n* = 8), the (**A**) mean time to peak (ms), (**B**) mean time to 50% relaxation (ms) and (**C**) mean time to 90% relaxation (ms) did not change compared to baseline. A similar response was documented for the BI-change group (*n* = 12) parameters: (**D**) mean time to peak (ms), (**E**) mean time to 50% relaxation (ms) and (**F**) mean time to 90% relaxation (ms). Data are presented as mean ± SEM.

**Figure 4 cells-12-01881-f004:**
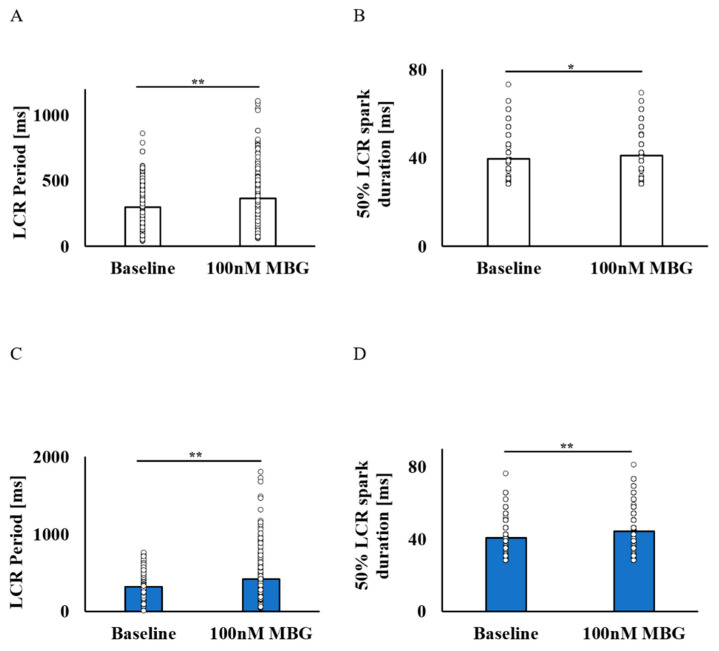
The effect of 100 nM MBG on rabbit SANC local Ca^2+^ release parameters. Change from baseline LCR parameters after administration of 100 nM MBG to cells with no change in BI. (**A**) Mean LCR period (ms) (*n* = 230) and (**B**) mean 50% LCR spark duration (ms) (*n* = 239) were increased. Change from baseline LCR parameters after administration of 100 nM MBG to cells with change in BI. (**C**) Mean LCR period (ms) (*n* = 291) and (**D**) mean 50% LCR spark duration (ms) (*n* = 305) were prolonged. Data are presented as mean ± SEM. * *p* ≤ 0.05, ** *p* ≤ 0.01.

**Figure 5 cells-12-01881-f005:**
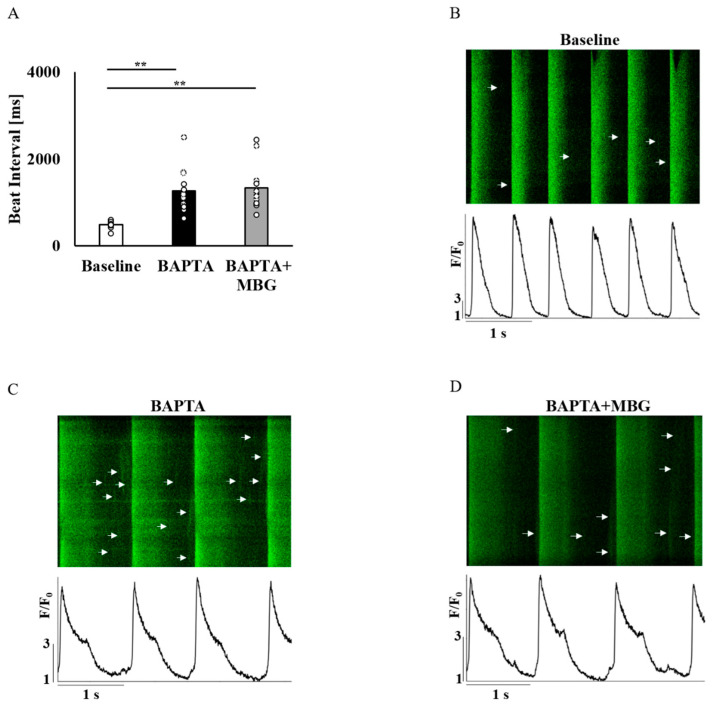
The effect of 10 µM BAPTA and 100 nM MBG on rabbit SANC global Ca^2+^ parameters. (**A**) The effect of coadministration of 10 µM BAPTA and 100 nM MBG on SANCs’ BI (*n* = 12). Representative example of (**B**) untreated SANC and of the effect of (**C**) BAPTA alone and (**D**) combined treatment of 10 µM BAPTA and 100 nM MBG on Ca^2+^ transients in SANCs. Data are presented as mean ± SEM. ** *p* ≤ 0.01.

**Figure 6 cells-12-01881-f006:**
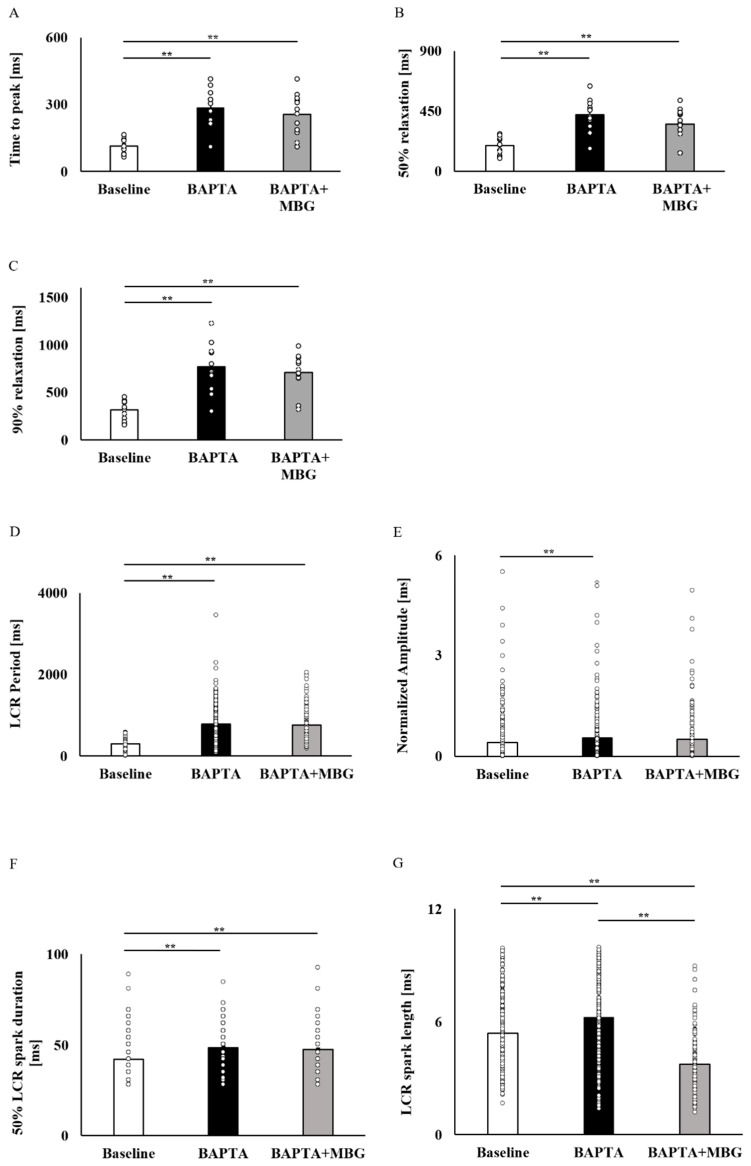
Elimination of the selective effect of MBG by Ca^2+^ chelation. Global parameters of SANCs treated with a combination of 10 µM BAPTA and 100 nM MBG. (**A**) Mean time to peak (ms) (*n* = 12), (**B**) mean time to 50% relaxation (ms) (*n* = 12) and (**C**) mean time to 90% relaxation (ms) (*n* = 12) did not change in comparison to cells treated with 10 µM BAPTA alone. Local Ca^2+^ release parameters of SANCs treated with a combination of 10 µM BAPTA and 100 nM MBG. (**D**) Mean LCR period (ms) (n_control_ = 276, n_10µM BAPTA_ = 212, n_10µM BAPTA+100nM MBG_ = 162), (**E**) mean LCR amplitude (*n*.U) (n_control_ = 349, n_10µM BAPTA_ = 218, n_10µM BAPTA+100nM MBG_ = 167) and (**F**) mean 50% LCR spark duration (ms) (n_control_ = 349, n_10µM BAPTA_ = 218, n_10µM BAPTA+100nM MBG_ = 167) was not different between cells subjected to combined treatment and cells treated with BAPTA alone. (**G**) Mean LCR spark length (µm) (n_control_ = 349, n_10µM BAPTA_ = 218, n_10 µM BAPTA+100nM MBG_ = 167) was lower in cells subjected to combined treatment than in cells treated with BAPTA alone. Data are presented as mean ± SEM. ** *p* ≤ 0.01.

## Data Availability

Data are available upon request.

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
