# Peer review of "Ca2+-Driven Selectivity of the Effect of the Cardiotonic Steroid Marinobufagenin on Rabbit Sinoatrial Node Function"

_cells, 2023, doi:10.3390/cells12141881_

Round 1

Reviewer 1 Report

In their MS the authors study the effect of  MBG the  Na/K ATPase blocker on SANC and then determine of the resulting rise in Ca2+ changes on the LCR as well as a beat interval. They found that indeed LCR is modulated and this effect can be reversed by intracellular Ca2+ chelation by BAPTA

this is an interesting study of important physiological implications 

the following issues should be addressed 

the claim here is that the Ca2+ rise is triggered by the reversal of NCX. The authors should show this by blocking NCK using KBR that particularly works on the reverse mode of the exchanger and determine if it affects LCR and BI 

2) the authors should present  the destruction of their data points in their bar graphs

Author Response

General comment

In their MS the authors study the effect of MBG the Na/K ATPase blocker on SANC and then determine of the resulting rise in Ca2+ changes on the LCR as well as a beat interval. They found that indeed LCR is modulated and this effect can be reversed by intracellular Ca2+ chelation by BAPTA. This is an interesting study of important physiological implication.  

We thank the reviewer for this feedback and for acknowledging the importance of the problem we aimed to address in the study.

Issue1

The claim here is that the Ca2+ rise is triggered by the reversal of NCX. The authors should show this by blocking NCK using KBR that particularly works on the reverse mode of the exchanger and determine if it affects LCR and BI 

Unfortunately, KB-R7943 is not specific and partially inhibits L-type calcium current (Cheng et al. Cell Calcium 2011). Inhibition of L-type current affects the SR Ca2+ load (Lyashkov et al. Biophysical Journal 2018) and thus other Ca2+-related mechanisms. Thus, addition of MBG will not yield conclusive results. Moreover, KB-R7943 slowed and then fully terminated spontaneous beating and thus LCR and BI could not be measured.

Because no evidence exists that Ca2+ rise is triggered by the reversal of NCX, we revised the discussion to read: “Based on the assumption that Na+-K+ pump blockade affects NCX, above a certain point, an increase in intracellular Ca2+ levels and Na+ leads to a decrease in NCX current and subsequently to a delay in action potential ignition. A prolonged diastolic depolarization phase leads to a reduction in the amount of Ca2+ entering the cell per time unit, and consequently to a reduction in LCR release. In parallel, prolonged diastolic depolarization leads to a reduction in the funny current per time unit, resulting in reduced clock coupling, leading to a further reduction in Ca2+ activity, and in phosphorylation activity [13,14]. Thus, the MBG effect on SANC BI is rooted in a change in the degree of clock coupling”

Issue2

The authors should present the destruction of their data points in their bar graphs

We thank the reviewer for this suggestion and revised Figs. 4 and 6; now all bar graphs include the data points.

Reviewer 2 Report

The manuscript of Segal and Yaniv showed that MBG, an endogenous cardiotonic steroid (CTS) in toads and possibly in humans, has selective effects on beat interval and spontaneous local calcium mobilization in a population of rabbit sinoatrial node cells, comparable to a previous study using an exogenous CTS (digoxigenin). The study is straightforward and should be considered for publication, but some comments are important.

- The authors correctly describe MBG as an endogenous mammalian steroid, but the concentrations they used are far above putative endogenous titers of the hormone. So, it is misleading. They should use the lower concentrations they consider in “Study limitation” to minimally link the in vitro translational effect (1 nM?);

- It is well known that CTS evoke signaling cascades mediated by Na/K-ATPase and dependent on Src. In order to clarify whether these protein-protein interactions are relevant to the effects they observe or it is only mediated by ionic (NCX) intervention, the experiments should be done with a Src and also a NCX inhibitor;

Minor comments

- ln. 79: Fluo, not Flou;

- ln. 100 and thereafter: concentration, not dose;

- Figs. 3 and 4: I would recommend that you show "paired" graphs (that is, for instance, in Fig. 3: A/D first, B/E middle, and C/F last).

Author Response

General comment

The manuscript of Segal and Yaniv showed that MBG, an endogenous cardiotonic steroid (CTS) in toads and possibly in humans, has selective effects on beat interval and spontaneous local calcium mobilization in a population of rabbit sinoatrial node cells, comparable to a previous study using an exogenous CTS (digoxigenin). The study is straightforward and should be considered for publication, but some comments are important.

We thank the reviewer for this feedback and for acknowledging the importance of the problem we aimed to address in the study. Below, please find our responses to each of your comments.

Major 1

The authors correctly describe MBG as an endogenous mammalian steroid, but the concentrations they used are far above putative endogenous titers of the hormone. So, it is misleading. They should use the lower concentrations they consider in “Study limitation” to minimally link the in vitro translational effect (1 nM?)

We added the following paragraph to the study limitation section: “The concentration used here is difficult to translate to clinically relevant concentrations. The fact that MBG does not affect the heart rate in vivo implies that the putative endogenous titers of the hormone are lower.”

Major 2

It is well known that CTS evoke signaling cascades mediated by Na/K-ATPase and dependent on Src. In order to clarify whether these protein-protein interactions are relevant to the effects they observe or it is only mediated by ionic (NCX) intervention, the experiments should be done with a Src and also a NCX inhibitor;

Blocking Src activity using PP2 leads to irregular spontaneous beat interval. Thus, it is challenging to measure the effect of MBG on further prolongation of BI. 

Unfortunately, KB-R7943 is not specific and partially inhibits L-type calcium current (Cheng et al. Cell Calcium 2011). Inhibition of L-type current affects the SR Ca2+ load (Lyashkov et al. Biophysical Journal 2018) and thus other Ca2+-related mechanisms. Thus, addition of MBG will not yield conclusive results. Moreover, KB-R7943 slowed and then fully terminated spontaneous beating and thus LCR and BI could not be measured.

Because no evidence exists that Ca2+ rise is triggered by the reversal of NCX, we revised the discussion to read: “Based on the assumption that Na+-K+ pump blockade affects NCX, above a certain point, an increase in intracellular Ca2+ levels and Na+ leads to a decrease in NCX current and subsequently to a delay in action potential ignition. A prolonged diastolic depolarization phase leads to a reduction in the amount of Ca2+ entering the cell per time unit, and consequently to a reduction in LCR release. In parallel, prolonged diastolic depolarization leads to a reduction in the funny current per time unit, resulting in reduced clock coupling, leading to a further reduction in Ca2+ activity, and in phosphorylation activity [13,14]. Thus, the MBG effect on SANC BI is rooted in a change in the degree of clock coupling”

Minor 1

  1. 79: Fluo, not Flou;

Thank you for noting this; the typo was corrected.

Minor 2

  1. 100 and thereafter: concentration, not dose;

The word dose was replaced by concentration.

Minor 3

Figs. 3 and 4: I would recommend that you show "paired" graphs (that is, for instance, in Fig. 3: A/D first, B/E middle, and C/F last).

We agree that paired graphs would be better. However, we first present the no BI change group and then the BI no changes and the panels had to be numbered in the order they were mentioned in the text. 

Round 2

Reviewer 1 Report

The authors addressed all the issues that I have raised 

Reviewer 2 Report

The authors answered all the comments. One small issue: In the last paragraph added by the authors, they should replace "clinically relevant concentrations" to "physiologically relevant concentrations".